# Muscle Cell Insulin Resistance Is Attenuated by Rosmarinic Acid: Elucidating the Mechanisms Involved

**DOI:** 10.3390/ijms24065094

**Published:** 2023-03-07

**Authors:** Danja J. Den Hartogh, Filip Vlavcheski, Evangelia Tsiani

**Affiliations:** 1Department of Health Sciences, Brock University, St. Catharines, ON L2S 3A1, Canada; 2Centre for Bone and Muscle Health, Brock University, St. Catharines, ON L2S 3A1, Canada

**Keywords:** muscle, insulin resistance, free fatty acid, palmitate, rosmarinic acid, IRS-1, GLUT4, AMPK

## Abstract

Obesity and elevated blood free fatty acid (FFA) levels lead to impaired insulin action causing insulin resistance in skeletal muscle, and contributing to the development of type 2 diabetes mellitus (T2DM). Mechanistically, insulin resistance is associated with increased serine phosphorylation of the insulin receptor substrate (IRS) mediated by serine/threonine kinases including mTOR and p70S6K. Evidence demonstrated that activation of the energy sensor AMP-activated protein kinase (AMPK) may be an attractive target to counteract insulin resistance. We reported previously that rosemary extract (RE) and the RE polyphenol carnosic acid (CA) activated AMPK and counteracted the FFA-induced insulin resistance in muscle cells. The effect of rosmarinic acid (RA), another polyphenolic constituent of RE, on FFA-induced muscle insulin resistance has never been examined and is the focus of the current study. Muscle cell (L6) exposure to FFA palmitate resulted in increased serine phosphorylation of IRS-1 and reduced insulin-mediated (i) Akt activation, (ii) GLUT4 glucose transporter translocation, and (iii) glucose uptake. Notably, RA treatment abolished these effects, and restored the insulin-stimulated glucose uptake. Palmitate treatment increased the phosphorylation/activation of mTOR and p70S6K, kinases known to be involved in insulin resistance and RA significantly reduced these effects. RA increased the phosphorylation of AMPK, even in the presence of palmitate. Our data indicate that RA has the potential to counteract the palmitate-induced insulin resistance in muscle cells, and further studies are required to explore its antidiabetic properties.

## 1. Introduction

Blood glucose homeostasis is tightly regulated by insulin. After postprandial glucose levels are increased, pancreatic β cells respond by releasing insulin into the bloodstream, where it is delivered to its target tissues. Specifically, insulin promotes the transport, utilization, and storage of glucose in skeletal muscle and adipose tissue [1,2], while inhibiting the endogenous production of glucose by the liver. These insulin actions are key to maintaining plasma glucose levels within a physiological range of 4–7 millimolar (mM).

Insulin initiates its action by binding to its receptor. This initiates tyrosine phosphorylation of the receptor and insulin receptor substrate (IRS-1), and activation of the lipid kinase phosphatidylinositol-3 kinase (PI3K) and the serine threonine kinase protein kinase B/Akt, resulting in increased translocation of the glucose transporter (GLUT4) from an intracellular pool to the plasma membrane and increased glucose uptake [3,4]. Impairments in the PI3K-Akt cascade can lead to the development of insulin resistance and type 2 diabetes mellitus (T2DM) [1,2,5].

Skeletal muscle accounts for roughly 70–80% of postprandial glucose uptake and is quantitatively the most important insulin target tissue. Therefore, skeletal muscle insulin resistance is a major contributor to decreased glucose tolerance and T2DM. Insulin resistance is strongly associated with obesity and increased plasma lipid levels. In vitro studies have shown that exposure of muscle cells to the free fatty acid (FFA) palmitate results in insulin resistance [6]. Additionally, evidence from in vivo animal studies demonstrated that lipid infusion [7,8] or increased plasma lipid levels due to a high-fat diet results in muscle insulin resistance [7,9]. Studies have shown that serine phosphorylation of IRS-1 (Ser^636/639^ and Ser^307^) results in an impaired insulin/PI3K/Akt signaling pathway and increased insulin resistance [6,10,11]. Signaling molecules including, mammalian target of rapamycin (mTOR) [12,13], ribosomal protein S6 kinase (p70S6K) [14,15], c-Jun N-terminal kinase (JNK) [16], and protein kinase C (PKCs) [17] act to increase the serine phosphorylation of IRS-1 [18].

The World Health Organization and the International Diabetes Federation (IDF) estimate that T2DM is a disease on the rise [19] and presents a huge economic burden on global health care systems. Different strategies to prevent and treat insulin resistance and T2DM do currently exist; however, they are lacking in efficacy and, therefore, there is a need for new preventative measures and targeted therapies.

Adenosine monophosphate (AMP)-activated protein kinase (AMPK) is a serine/threonine kinase that acts as a cellular energy sensor. AMPK is activated by an increased AMP/ATP ratio and/or via phosphorylation of its α-catalytic subunit by its upstream kinases, liver kinase B1 (LKB1), calmodulin-dependent protein kinase (CaMKKs), and transforming growth factor-β (TGF-β)-activated kinase 1 (TAK1) [20,21]. AMPK activity is driven by muscle contraction/exercise [21] and stimulated by several compounds, including the currently used antidiabetic drugs metformin [22] and thiazolidineones [23], and by various polyphenols/flavanols found in tea [24], red wine, citrus fruit, and cocoa [25], including resveratrol [26], naringenin [27], and cocoa flavanol [28], resulting in increased skeletal muscle glucose uptake. The utilization of AMPK activators has gained increasing attention as a novel pharmacological intervention for the prevention/treatment of T2DM and insulin resistance [21,29,30,31]. Chemicals found in plants/herbs that activate AMPK have attracted attention as potential diabetes treatment options.

Rosemary (*Rosmarinus officinalis* Lamiaceae) is an aromatic evergreen plant reported to have antioxidant [32,33], anticancer [18,20], and antidiabetic properties [34,35,36,37,38]. Rosemary extract (RE) contains different classes of polyphenols, including phenolic acids (rosmarinic acid; RA), flavonoids, and phenolic diterpenes (carnosic acid; CA and carnosol; COH) [39].

Previous studies by our group showed a significant increase in muscle glucose uptake and AMPK activation by RE [40], CA [41], RA [42], and COH [43] treatment. More importantly, treatment of L6 muscle cells with RE [44] and CA [45] attenuated the palmitate-induced insulin resistance.

Limited evidence from in vitro and in vivo models of insulin resistance indicate that RA may potentially be used to improve insulin sensitivity [46,47,48]. However, the mechanistic effects of RA on palmitate-induced insulin-resistant muscle cells remain to be elucidated.

In the present study, we examined the potential of RA to counteract palmitate-induced insulin resistance in muscle cells.

## 2. Results

### 2.1. The Palmitate-Induced Serine Phosphorylation of Insulin Receptor Substrate-1 (IRS-1) Is Prevented by Rosmarinic Acid Treatment

An increase in the serine phosphorylation of IRS-1 at residues Ser^307^ and Ser^636/639^ is linked to impaired PI3K/Akt signaling and insulin resistance, and for this reason we first examined the effect of RA on IRS-1. Exposure of L6 muscle cells to 0.2 mM palmitate for 16 h significantly increased IRS-1 phosphorylation at residues Ser^307^ and Ser^636/639^ (P: 134.7 ± 9.2% and 140.1 ± 7.2% of control, *p* < 0.05 and *p* < 0.01, respectively, Figure 1A,B). The palmitate-induced Ser^307^ and Ser^636/639^ phosphorylation of IRS-1 was abolished with RA treatment (RA + P: 58.1 ± 10.5% and 105.0 ± 7.8% of control, *p* < 0.01 and *p* < 0.05, respectively, Figure 1A,B). Treatment with RA alone reduced Ser^307^ phosphorylation of IRS-1 (RA: 60.8 ± 10.7% of control, *p* < 0.05, Figure 1A,B), but had no effect on basal Ser^636/639^ phosphorylation of IRS-1 (RA: 98.3 ± 11.3% of control, Figure 1A,B). Moreover, the total levels of IRS-1 were unaffected by any treatment (P: 108.5 ± 2.5%, RA: 99.7 ± 8.1%, RA + P: 129.0 ± 12.1% of control, Figure 1A,B).

### 2.2. The Insulin-Stimulated Akt Phosphorylation in Palmitate-Treated Myotubes Is Restored with Rosmarinic Acid

Akt phosphorylation/activation is a key step in the insulin signaling cascade, leading to increased glucose uptake by muscle cells, and is impaired in insulin resistance [49]. Therefore, we investigated the effect of RA on Akt. Treatment of L6 myotubes with 100 nM insulin for 30 min resulted in a significant increase in Akt Ser^473^ phosphorylation, an indicator of activation (I: 816.5 ± 109.87% of control, *p* < 0.01, Figure 2A,B). Exposure of the cells to palmitate impaired the insulin-stimulated phosphorylation of Akt (P + I: 159.7 ± 49.4% of control, *p* = 0.009, Figure 2A,B). However, in the presence of RA, insulin-stimulated Akt phosphorylation was restored (RA + P + I: 507.2 ± 67.17% of control, *p* < 0.01, Figure 2A,B). Palmitate alone had no effect on basal Akt phosphorylation (P: 71.0 ± 5.04% of control, Figure 2A,B). The total levels of Akt were not significantly changed by any of the treatments (I: 110.3 ± 31.5%, P: 106.9 ± 12.8%, P + I: 118.8 ± 29.6%, RA + P + I: 109.8 ± 33.4% of control, Figure 2A,B).

### 2.3. Rosmarinic Acid Restores Insulin-Stimulated GLUT4 Translocation to Plasma Membrane in Palmitate-Treated Myotubes

Skeletal muscle glucose uptake in response to insulin is driven by glucose transporter GLUT4 translocation from an intracellular pool to the plasma membrane and is mediated by upstream activation of the PI3K/Akt cascade. We examined the effects of our treatment on GLUT4 transporter translocation to the plasma membrane using L6 cells that overexpress an myc-labelled GLUT4 glucose transporter [50]. Acute stimulation of GLUT4myc overexpressing L6 myotubes with 100 nM insulin for 30 min resulted in a significant increase in GLUT4 plasma membrane levels (I: 193.0 ± 6.42% of control, *p* < 0.001, Figure 3). Palmitate impaired the insulin-stimulated GLUT4 plasma membrane levels (P + I: 131.4 ± 5.48% of control, Figure 3) while RA restored the insulin-stimulated GLUT4 plasma membrane levels (RA + P + I: 175.1 ± 9.26% of control, *p* < 0.01, Figure 3).

### 2.4. Rosmarinic Acid Restores the Insulin-Stimulated Glucose Uptake in Palmitate-Treated Myotubes

Next, we examined the effects of RA on muscle cell glucose uptake. Stimulation of L6 myotubes with 100 nM insulin for 30 min significantly increased glucose uptake (201 ± 1.21% of control, *p* < 0.0001, Figure 4). Exposure of the cells to 0.2 mM palmitate for 16 h almost abolished the insulin-stimulated glucose uptake (P + I: 119 ± 13.2% of control), indicating impaired insulin action. Most importantly, palmitate-treated cells exposed to 5 µM RA had significantly increased insulin-stimulated glucose uptake (RA + P + I: 184 ± 15.5% of control *p* < 0.001, Figure 4). Treatment with RA in the presence of palmitate did not have a significant effect on basal glucose uptake (RA + P: 124 ± 5.8% of control).

### 2.5. The Palmitate-Induced Phosphorylation/Activation of mTOR and p70S6K Is Prevented in the Presence of Rosmarinic Acid

mTOR and p70S6K are kinases implicated in serine phosphorylation of IRS-1 and are specifically known to phosphorylate Ser^307^ and Ser^636/639^, and therefore, we examined the effects of palmitate on mTOR and p70S6K phosphorylation/activation and expression. Exposure of the cells to 0.2 mM palmitate for 16 h significantly increased mTOR Ser^2448^ and p70S6K Thr^389^ phosphorylation (P: 174.6 ± 15.6% and 572.7 ± 57.8% of control, respectively, *p* < 0.01, Figure 5A–D). Treatment with RA alone did not affect the basal mTOR (RA: 91.8 ± 8.3% of control, Figure 5A–D) or p70S6K (RA: 203.9 ± 60.9% of control, Figure 5A–D) phosphorylation levels. However, RA treatment significantly prevented the palmitate-induced phosphorylation of mTOR and p70S6K (RA + P: 105.8 ± 4.35% and 247.2 ± 54.02% of control, respectively, *p* < 0.05, Figure 5A–D). The total levels of mTOR and p70S6K were not significantly changed by any treatment.

### 2.6. Rosmarinic Acid Increases the Phosphorylation of AMPK, ACC, and Raptor

Previous studies by our group showed that RE and RE polyphenols increased muscle cell glucose uptake via activation of AMPK [40,41,42,43]. Here, we investigated the effect of RA on AMPK and its downstream targets ACC and Raptor.

The phosphorylation of AMPK at Thr^172^ was significantly increased in cells treated with 5 µM RA (RA: 252.8 ± 36.8% of control, *p* < 0.05, Figure 6A,B). We also examined phosphorylation of ACC, a direct target of activated AMPK, which has been established as a marker of AMPK activation. RA increased the phosphorylation of ACC (RA: 170.6 ± 18.6% of control, *p* < 0.01, Figure 6C,D). Most importantly, RA increased the phosphorylation of AMPK and ACC even in the presence of 0.2 mM palmitate (RA + P: 229.4 ± 34.3% and 178.9 ± 25.3% of control, *p* < 0.05 and *p* < 0.01, respectively, Figure 6A–D). Treatment with palmitate alone had no significant effect on phosphorylated AMPK and ACC levels (P: 87% and 74% of control, respectively, Figure 6A–D). Furthermore, the total levels of AMPK (P: 121 ± 38%, RA: 103 ± 22%, RA + P: 108 ± 24% of control, Figure 6A,B), and ACC (P: 104 ± 14%, RA: 93 ± 8%, RA + P: 99 ± 12% of control, Figure 6C,D) were not affected by any treatment.

The activity of mTOR is influenced by the regulatory-associated protein of the mammalian target of rapamycin (Raptor), and the phosphorylation of Raptor on Ser^792^ results in inhibition of mTOR [51,52,53]. AMPK activation leads to the direct phosphorylation of Raptor. Treatment with 5 μM RA increased the phosphorylation of Raptor (RA: 153 ± 5.7% of control, *p* < 0.01, Figure 6E,F). Most importantly, RA increased the phosphorylation of Raptor even in the presence of 0.2 mM palmitate (RA + P: 155 ± 7.9% of control, *p* < 0.05, Figure 6E,F). Treatment with palmitate alone had no effect on the phosphorylation of Raptor (P: 102 ± 8.1% of control). Furthermore, the total levels of Raptor were not affected by any treatment (P: 96 ± 4%, RA: 94 ± 2%, RA + P: 93 ± 4% of control, Figure 6E,F).

### 2.7. AMPK Inhibition Reverses the Effects of Rosmarinic Acid on Palmitate-Induced Phosphorylation of Raptor, mTOR, and p70S6K

In an attempt to elucidate the mechanism of action of RA, we performed additional experiments utilizing compound C (CC), a specific AMPK inhibitor. The phosphorylation of Raptor at Ser^792^ was significantly increased in cells treated with 5 µM RA in the presence of 0.2 mM palmitate (RA + P: 193.63 ± 23.7% of control, *p* < 0.05, Figure 7A,B), and importantly, pretreatment of the cells with CC abolished this response (RA + P + CC: 87.2 ± 16.04% control, *p* < 0.05, Figure 7A,B). Furthermore, in the presence of CC, the effect of RA on suppressing the palmitate-induced mTOR and p70S6Kphosphorylation/activation (P: 244.2 ± 23.3% and 259.3 ± 34.8% of control, *p* < 0.001 and *p* < 0.05, respectively, Figure 7C–F), (RA + P: 121.6 ± 12.8% and 114.1 ± 14.7% of control, *p* < 0.01 and *p* < 0.05, respectively, Figure 7C–F) was abolished (RA + P + CC: 241.5 ± 30.1% and 272.2 ± 14.9% control, *p* < 0.05 and *p* < 0.01, respectively, Figure 7C–F).

## 3. Discussion

Obesity and elevated FFA levels are strong indicators of insulin resistance and are major risk factors for T2DM [18], a disease affecting millions of people globally. Current drugs used to treat insulin resistance and T2DM manifest negative side effects, driving the search for natural, plant-derived compounds with the potential to counteract insulin resistance. In the current study, we examined the potential of the plant-derived compound

RA to counteract the palmitate-induced insulin resistance in muscle cells. Exposure of L6 muscle cells to palmitate, a saturated FFA, to simulate the in vivo scenario of elevated FFA plasma levels often seen in obesity, dramatically reduced the plasma membrane levels of GLUT4 and insulin-stimulated glucose uptake, indicating insulin resistance (Figure 7). These findings are in agreement with previous data from our lab [44,45,54] and others [6,47,55,56]. Importantly, in the presence of RA, the insulin-stimulated plasma membrane GLUT4 levels and glucose uptake were restored to levels comparable to those achieved with insulin stimulation alone. Only one other study examined the effects of RA on L6 muscle cells. Jayanthy et al. found that treatment of L6 muscle cells with RA (20 µM) and palmitate (0.3 mM) for 24 h increased GLUT4 plasma membrane levels and glucose uptake [47]. Unfortunately, the study by Jayanthy et al. did not examine the effects of RA on insulin-induced responses. Although, in our study, we found that RA restored insulin responsiveness, we did not find any significant effect of RA on basal GLUT4 and glucose uptake. The differences between our study and that by Jayanthy et al. may be due to the different RA (5 vs. 20 µM) and/or palmitate (0.2 vs. 0.3 mM) concentrations used.

Additionally, our study found that exposure of L6 muscle cells to palmitate markedly reduced the insulin-stimulated Akt phosphorylation and is in agreement with other in vitro studies using L6 [57] and C2C12 [58] muscle cells as well as in vivo studies showing reduced levels of Akt phosphorylation in soleus muscle harvested from mice fed a high fat diet (HFD) [59]. Remarkably, treatment with RA restored the insulin-stimulated phosphorylation of Akt, indicating that RA, similarly to metformin [60], counteracts the harmful effects of palmitate.

In addition, our study found that treatment of L6 cells with palmitate increased the serine phosphorylation of IRS-1. This finding is in agreement with other studies indicating increased Ser^307^ and Ser^636/639^ phosphorylation of IRS-1 in the presence of palmitate in L6 [56] and C2C12 [61] cells. Increased phosphorylation of the serine residues of IRS-1 results in reduced downstream PI3K-Akt signaling and glucose uptake [62,63]. Importantly, our studies show that RA blocked the palmitate-induced serine phosphorylation of IRS-1, an effect that is similar to that of metformin [64]. These data are in agreement with the study by Jayanthy et al. that showed a reduction in the palmitate-induced Ser^307^ phosphorylation of IRS-1 via RA treatment of L6 muscle cells [47].

Furthermore, exposure of L6 myotubes to palmitate considerably augmented the phosphorylation of mTOR and its downstream effector p70S6K, and most importantly, treatment with RA attenuated these effects of palmitate (Figure 8). While it has already been established by other studies that palmitate treatment results in increased mTOR phosphorylation in L6 [65] and C2C12 cells [66], and in muscle tissue obtained from animals fed an HFD [65,67], this study is the first to report that RA has the potential to block these effects and act in a similar fashion as the mTOR inhibitor rapamycin [68] and metformin [69]. Importantly, studies have shown that the Ser^307^ and Ser^636/639^ phosphorylation of IRS-1 is mediated by mTOR and its downstream target p706SK to reduce PI3K/Akt signaling and glucose uptake [62,70]. The role of mTOR/p706SK signaling has been confirmed by many other groups as a critical mechanism involved in the induction of insulin resistance in insulin-sensitive tissues (muscle, fat and liver) [71,72,73,74]. Several studies have shown that activated mTOR causes phosphorylation of the growth factor receptor-binding protein 10 (Grb10) at Ser476, which in turn binds to the phosphorylated tyrosine residues of the insulin receptor and inhibits its tyrosine kinase activity, resulting in reduced downstream PI3K-Akt signaling [75,75,76]. Additionally, co-immunoprecipitation studies revealed that Grb10 was found to bind to the regulatory p85 subunit of PI3K, indicating that Grb10 directly associates with PI3K and reduces the PI3K catalytic activity, resulting in impaired insulin action in L6 myotubes [75]. Furthermore, overexpression of Grb10 inhibits the interaction of the insulin receptor with PI3K, thus reducing insulin signaling and causing insulin resistance [76,77]. Although we did not examine the effects of RA on Grb10, it is possible that RA treatment reduces Grb10 levels.

Furthermore, our present study shows that RA markedly increased the phosphorylation/activation of AMPK even in the presence of palmitate, indicating that the effects of RA are similar to those of metformin, which also activates AMPK in the presence of palmitate in L6 and C2C12 myotubes [60,69]. Activation of AMPK directly inhibits mTOR activity by phosphorylating Raptor at Ser^722/792^ [78,79]. Our study indicates an activation/phosphorylation of Raptor with RA treatment, and that the RA-induced activation of AMPK may be the reason for the inhibition of mTOR and its downstream target p706SK (Figure 8). Indeed, pretreatment of the cells with the specific AMPK inhibitor compound C (CC) abolished the effects of RA on palmitate-induced responses. These data indicate that activation of AMPK plays a major role in the RA-induced effects.

A small number of studies have investigated the antidiabetic effects of RA in vivo. In HFD-induced diabetic rats, the intraperitoneal administration of RA (200 mg/kg/day for 28 days) dose-dependently ameliorated hyperglycemia and increased insulin sensitivity assessed by the oral glucose tolerance test (OGTT). These effects were associated with reduced hepatic PEPCK protein expression and increased skeletal muscle GLUT4 protein levels [46]. Additionally, treatment of STZ-induced diabetic rats with RA resulted in amelioration of hyperglycemia [46]. Another study found that RA administration through oral gavage (100 mg/kg/day for 30 days) improved glucose homeostasis and significantly increased AMPK phosphorylation/activation and mitochondrial biogenesis/activity in the skeletal muscles of STZ-HFD-induced insulin-resistant rats [47]. These studies demonstrate that RA exhibits anti-hyperglycemic and antidiabetic properties in vivo. However, there are currently no studies that elucidate the mechanisms involved in the effects of RA. The present study found that RA prevented the palmitate-induced phosphorylation/activation of mTOR and p70S6K and restored insulin-stimulated Akt phosphorylation, GLUT4 glucose transporter translocation to the plasma membrane, and glucose uptake (Figure 8).

## 4. Materials and Methods

### 4.1. Materials

The following materials were purchased from Sigma Life Sciences (St. Louis, MO, USA): bovine serum albumin (BSA), compound C (CC), cytochalasin B, dimethyl sulfoxide (DMSO), fetal bovine serum (FBS), palmitic acid, and rosmarinic acid. Trypan blue solution 0.4% and material necessary for cell culture were purchased from GIBCO Life Technologies (Burlington, ON, Canada). [3H]-2-deoxy-D-glucose was purchased from PerkinElmer (Boston, MA, USA). Antibodies including phospho and total ACC (CAT 3661 and 3662, respectively), Akt (CAT 9271 and 9272, respectively), AMPK (CAT 2531 and 2532, respectively), HRP-conjugated anti-rabbit (CAT 7074), IRS-1 (CAT 2381, 2388, and 2382, respectively), mTOR (CAT 2971 and 2972, respectively), and p70S6K (CAT 9205 and 9202, respectively) were purchased from New England BioLabs (NEB) (Missisauga, ON, Canada). Insulin (Humulin R) was obtained from Eli Lilly (Indianapolis, IN, USA). Materials for Western blotting and Bradford protein assay reagent were purchased from BioRad (Hercules, CA, USA).

### 4.2. Preparation of Palmitate Stock Solution

Stock palmitate solution was prepared by conjugating palmitic acid with fatty-acid-free BSA as previously reported by us [44,45,54] and others [6]. In short, palmitic acid was dissolved in 0.1 N NaOH and diluted in (45–50 °C) prewarmed 9.7% (*w*/*v*) BSA solution. A stock solution of 8 mM palmitate with a final molar ratio of free palmitate/BSA of 6:1 was prepared and kept at −80 °C.

### 4.3. Cell Culture, Treatment, and Glucose Uptake

All experiments utilized L6 rat skeletal muscle cells. Undifferentiated myoblasts were grown in α-Minimum Essential Medium (MEM) media containing 10% *v*/*v* FBS until 80% confluency and differentiated into myotubes in α-MEM media containing 2% *v*/*v* FBS, as previously established [5,26]. Fully differentiated myotubes were achieved approximately 6 to 7 days after seeding. All treatments were performed using serum-free α-MEM media containing 0% *v*/*v* FBS, followed by exposure to palmitate (0.2 mM) in the absence or presence of RA (5 µM) for 16 h followed by treatment without or with insulin (100 nM) for 0.5 h. Following treatment, the cells were rinsed with HEPES-buffered saline (HBS) and exposed to [3H]-2-deoxy-D-glucose (10 µM) for 10 min to measure cellular glucose uptake, as previously described [26,80]. Non-specific glucose uptake was measured in the presence of cytochalasin B (10 µM) and was subtracted from the total to obtain the specific glucose uptake. At the end of the glucose uptake assay, the cells were rinsed with 0.9% NaCl and lysed using a 0.05 N NaOH solution. A liquid scintillation counter was used to measure the radioactivity, and the Bradford assay was used to determine the protein content.

### 4.4. GLUT4myc Translocation Assay

Fully differentiated L6 GLUT4myc-overexpressing myotubes, grown in 24-well plates, were treated and fixed using 3% paraformaldehyde dissolved in PBS for 10 min at 4 °C. The fixed cells were then rinsed and incubated with PBS containing 1% glycine for 10 min, followed by blocking with 10% goat serum containing PBS for 15 min. The cells were then exposed to blocking buffer containing primary anti-myc antibody (1 h, 1:500), followed by incubation with blocking buffer containing HRP-conjugated donkey anti-mouse antibodies (45 min, 1:1000) at 4 °C. The cells were washed extensively with PBS and were incubated with O-phenylenediamine dihydrochloride (OPD) reagent at room temperature and protected from light for 30 min. The reaction was stopped using 3 N HCL solution, and the supernatant was collected and visualized at 492 nm. The OPD reagent is a substrate for HRP and produces a yellow product that can be visualized using an absorbance plate reader (Synergy HT, BioTek Instruments, Winooski, VT, USA). The intensity of the color produced is proportionate to the levels of GLUT4myc detected in the plasma membrane.

### 4.5. Immunoprecipitation

Whole-cell lysates (150 µg) were incubated with IRS1 antibody (1:50 volume ratio) conjugated to SureBeads^TM^ Protein G Magnetic beads (Biorad; Hercules, CA, USA) for 1 h at room temperature. The lysates–IRS-1–beads complex was collected by microcentrifugation and washed three times with PBS + 0.1% Tween-20. Protein was eluted with glycine (20 mM, pH 2.0) solution for 5 min at room temperature and neutralized with PBS (1 M, pH 7.4) at 10% eluent volume. A 3× sodium dodecyl sulfate (SDS) sample buffer was added to eluted protein and boiled for 5 min.

### 4.6. Immunoblotting

After treatment, L6 myotubes were rinsed twice with pre-chilled (4 °C) PBS solution and the cells were lysed using ice-cold lysis buffer containing ethylene glycol-bis β-aminoethyl ether/egtazic acid (EGTA), 1 mM ethylenediaminetetraacetic acid (EDTA), 1mM sodium orthovanadate (Na_3_VO_4_), 1 mM p-glycerolphosphate, 20 mM Tris (pH 7.5), 1% Triton X-100, 1 mM, 150 mM NaCI, 1 µg/mL leupeptin, 2.5 mM sodium pyrophosphate and 1 mM phenylmethylsulfonyl fluoride (PMSF) and were stored at −20 °C. A 5% B-mercaptoethanol containing 3x SDS buffer was added and the samples were boiled for 5 min. The proteins were separated using SDS-polyacrylamide gel electrophoresis (SDS-PAGE) and transferred to a polyvinylidene fluoride (PVDF) membrane followed by blocking with Tris-buffered saline containing 5% (*w*/*v*) dry milk powder and incubation with the primary antibody overnight at (4 °C). To detect the primary antibody HRP-conjugated anti-rabbit secondary antibodies were used followed by exposure to LumGLOW reagent. The blots were visualized using ChemiDoc, imaging system (BioRad, Hercules, CA, USA).

### 4.7. Statistical Analysis

Statistical analysis was completed using GraphPad Prism software 5.3 manufactured by Graphpad Software Inc. (La Jolla, CA, USA). The data from several experiments were pooled and presented as mean ± standard error (SE). The means of all the groups were obtained and compared to the control group using one-way analysis of variance (ANOVA), which was followed by Tukey’s post hoc test for multiple comparisons.

## 5. Conclusions

The prevalence of T2DM is constantly increasing, and according to the International Diabetes Federation, it is expected to affect 420 million people worldwide by the year 2040 [19]. In addition, insulin resistance and T2DM are highly correlated with the development of other pathological states, including cardiovascular disease and cancer [18]. As a result, new strategies to aid in the prevention and management of T2DM will provide huge benefits to our society. As previously indicated, increased levels of FFA and obesity mediates insulin resistance in muscle cells. The present study has shown that the exposure of muscle cells to the FFA palmitate, as a way to mimic the elevated FFA levels seen in obesity, induced insulin resistance. Palmitate exposure to L6 muscle cells increased the phosphorylation of mTOR and p70S6K, while insulin-stimulated Akt phosphorylation and the insulin-stimulated glucose uptake and GLUT4 translocation were significantly reduced. Importantly, these effects of palmitate were attenuated by RA treatment, and insulin-stimulated glucose uptake was restored. In addition, RA increased the phosphorylation/activation of the energy sensor AMPK, an attractive target to counteract insulin resistance and T2DM. Our study is the first to show that RA has the potential to counteract palmitate-induced muscle cell insulin resistance, and further studies are required to explore its antidiabetic properties and to elucidate the exact cellular mechanisms involved.

## Figures and Tables

**Figure 1 ijms-24-05094-f001:**
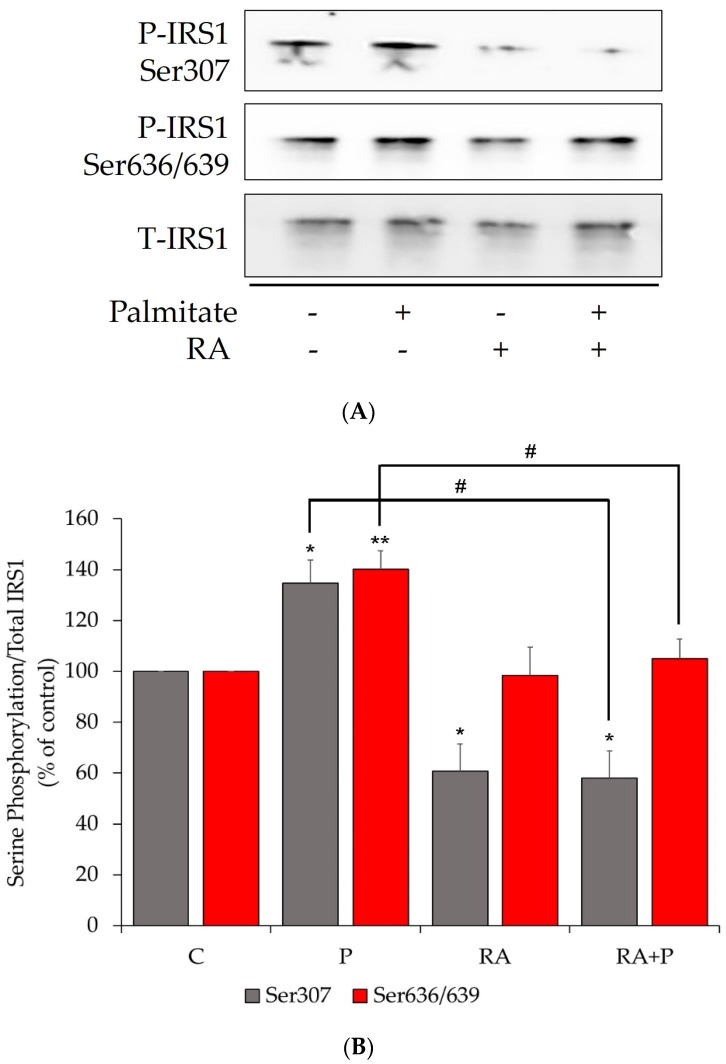
Rosmarinic acid attenuates the palmitate-induced serine (Ser307 and Ser636/639) phosphorylation of muscle cell IRS-1. Fully differentiated L6 myotubes were treated without (control, C) or with 0.2 mM palmitate (P) for 16 h in the absence or the presence of 5 μM rosmarinic acid (RA). After treatment, the cells were lysed, and IRS-1 was immunoprecipitated (IP). The IPs were immunoblotted for phosphorylated Ser^307^ and Ser^636/639^ or total IRS-1. Representative immunoblots are shown (**A**). The intensity of the bands (arbitrary units) was measured using Image J software and expressed as percent of control (**B**). The data are the mean ± SE of three separate experiments (* *p* < 0.05, ** *p* < 0.01 vs. control, # *p* < 0.05 vs. palmitate alone).

**Figure 2 ijms-24-05094-f002:**
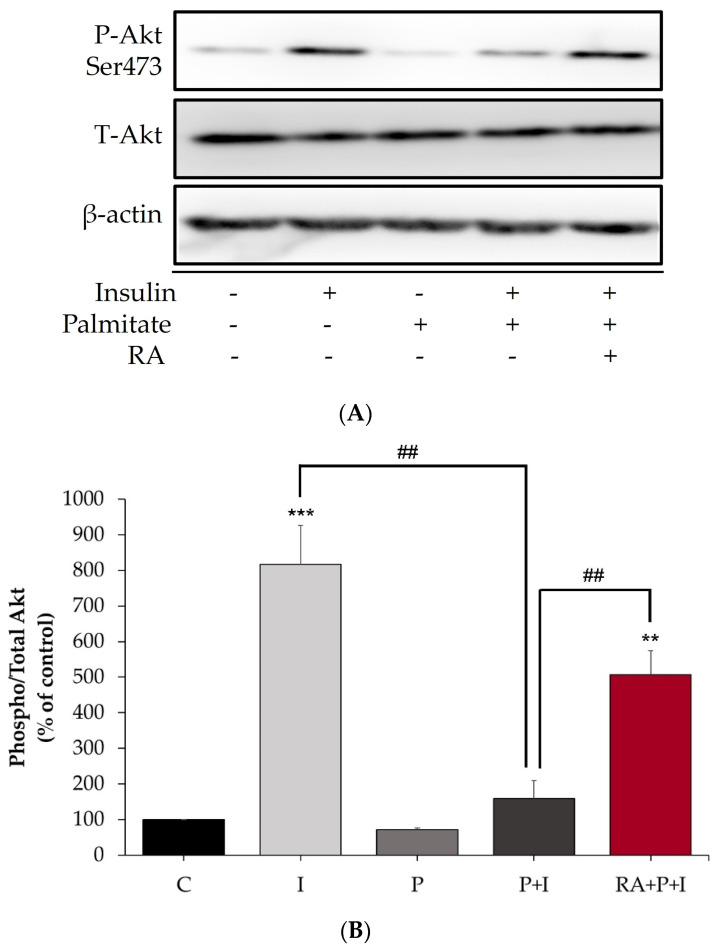
Rosmarinic acid abolishes the deleterious effects of palmitate on insulin-stimulated Akt. Fully differentiated L6 myotubes were treated without (control, C) or with 0.2 mM palmitate (P) for 16 h in the absence or the presence of 5 μM rosmarinic acid (RA) followed by stimulation without or with 100 nM insulin (I) for 30 min. After treatment, the cells were lysed, and SDS-PAGE was performed, followed by immunoblotting with specific antibodies that recognize phosphorylated Ser^473^ or total Akt. Representative immunoblots are shown (**A**). The intensity of the bands (arbitrary units) was measured using Image J software and expressed as percent of control (**B**) The data are the mean ± SE of four separate experiments (** *p* < 0.01, *** *p* < 0.001 vs. control, ## *p* < 0.01 as indicated).

**Figure 3 ijms-24-05094-f003:**
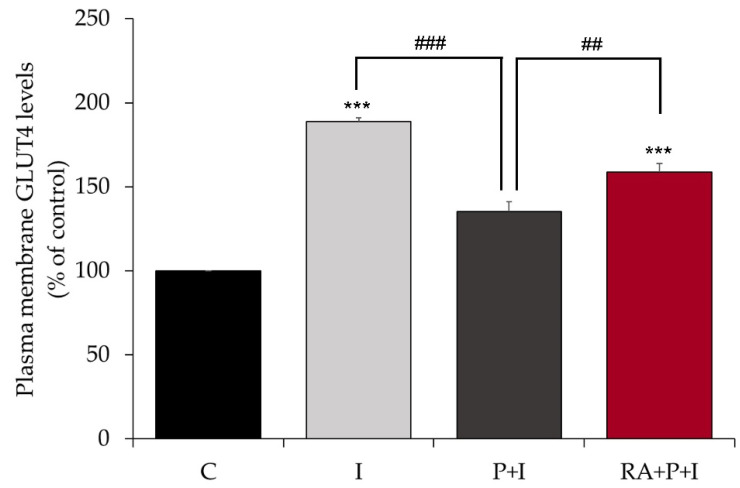
Rosmarinic acid abolishes the deleterious effects of palmitate on insulin-stimulated GLUT4 translocation. GLUT4myc overexpressing L6 myotubes were treated without (control, C) or with 0.2 mM palmitate (P) for 16 h in the absence or the presence of 5 μM rosmarinic acid (RA) followed by stimulation without or with 100 nM insulin (I) for 30 min and GLUT4 glucose transporter plasma membrane levels were measured. Results are the mean ± SE of three independent experiments performed in triplicate (*** *p* < 0.001 vs. control, ## *p* < 0.01, ### *p* < 0.001 as indicated).

**Figure 4 ijms-24-05094-f004:**
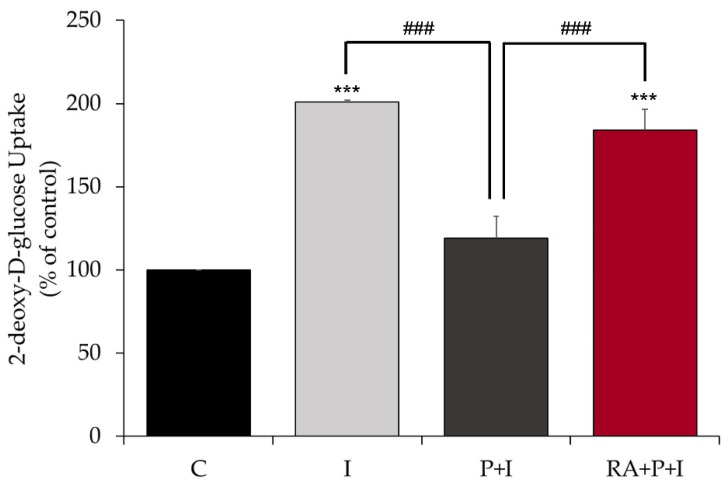
Rosmarinic acid restores the insulin-stimulated glucose uptake in palmitate-treated muscle cells. Fully differentiated L6 myotubes were treated without (control, C) or with 0.2 mM palmitate (P) for 16 h in the absence or the presence of 5 µM rosmarinic acid (RA) followed by stimulation without or with 100 nM insulin (I) for 30 min and [3H]-2-deoxy-D-glucose uptake measurements. The results are the mean ± standard error (SE) of five independent experiments, expressed as percent of control (*** *p* < 0.001 vs. control, ### *p* < 0.001 vs. insulin alone).

**Figure 5 ijms-24-05094-f005:**
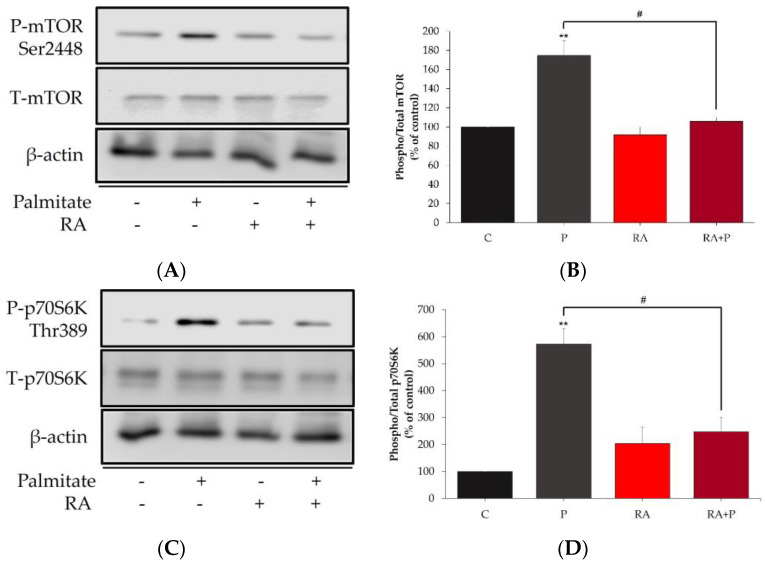
Rosmarinic acid prevents palmitate-induced mTOR (**A**,**B**) and p70S6K (C,D) phosphorylation/activation. Fully differentiated myotubes were treated without (control, C) or with 0.2 mM palmitate (P) for 16 h in the absence or the presence of 5 μM rosmarinic acid (RA). After treatment, the cells were lysed, and SDS-PAGE was performed, followed by immunoblotting with specific antibodies that recognize phosphorylated Ser^2448^ or total mTOR (**A**,**B**), and phosphorylated Thr^389^ or total p70S6K (**C**,**D**). Representative immunoblots are shown (**A**,**C**), and the intensity of the bands (arbitrary units) was measured using Image J software and expressed as percent of control (**B**,**D**). The data are the mean ± SE of three separate experiments (** *p* < 0.01 vs. control, # *p* < 0.05 vs. palmitate alone).

**Figure 6 ijms-24-05094-f006:**
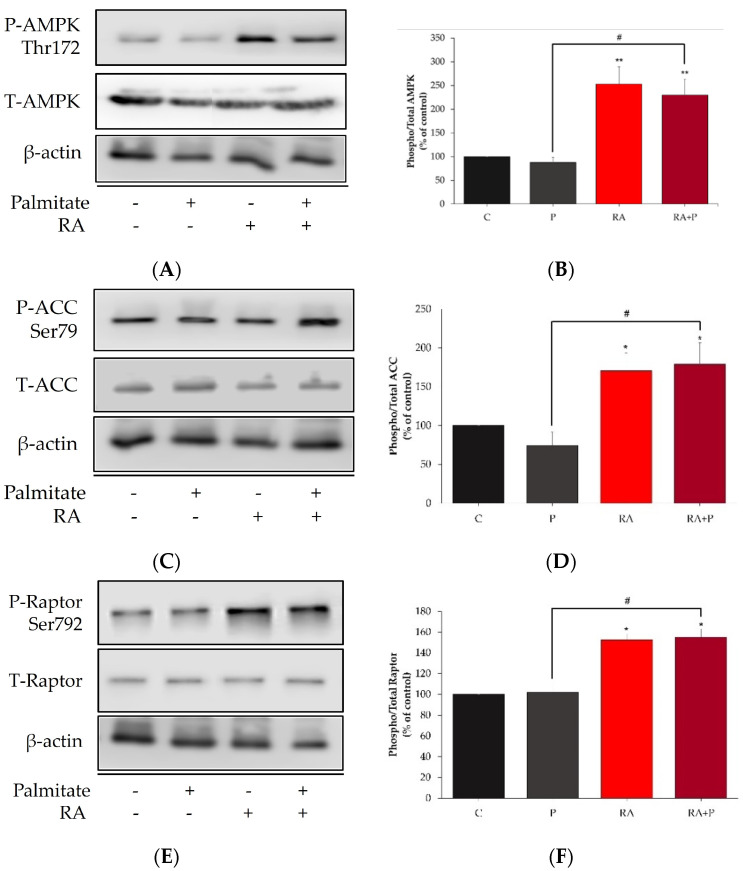
Rosmarinic acid phosphorylates AMPK (**A**,**B**), ACC (**C**,**D**), and Raptor (**E**,**F**). Fully differentiated myotubes were treated without (control, C) or with 0.2 mM palmitate (P) for 16 h in the absence or the presence of 5 μM rosmarinic acid (RA). After treatment, the cells were lysed, and SDS-PAGE was performed, followed by immunoblotting with specific antibodies that recognize phosphorylated Thr^172^ or total AMPK, phosphorylated Ser^79^ or total ACC, and phosphorylated Ser^792^ or total Raptor. Representative immunoblots are shown (**A**,**C**,**E**). The intensity of the bands (arbitrary units) was measured using Image J software and expressed as percent of control (**B**,**D**,**F**). The data are shown as the mean ± SE of three separate experiments (* *p* < 0.05, ** *p* < 0.01 vs. control, # *p* < 0.05 vs. palmitate alone).

**Figure 7 ijms-24-05094-f007:**
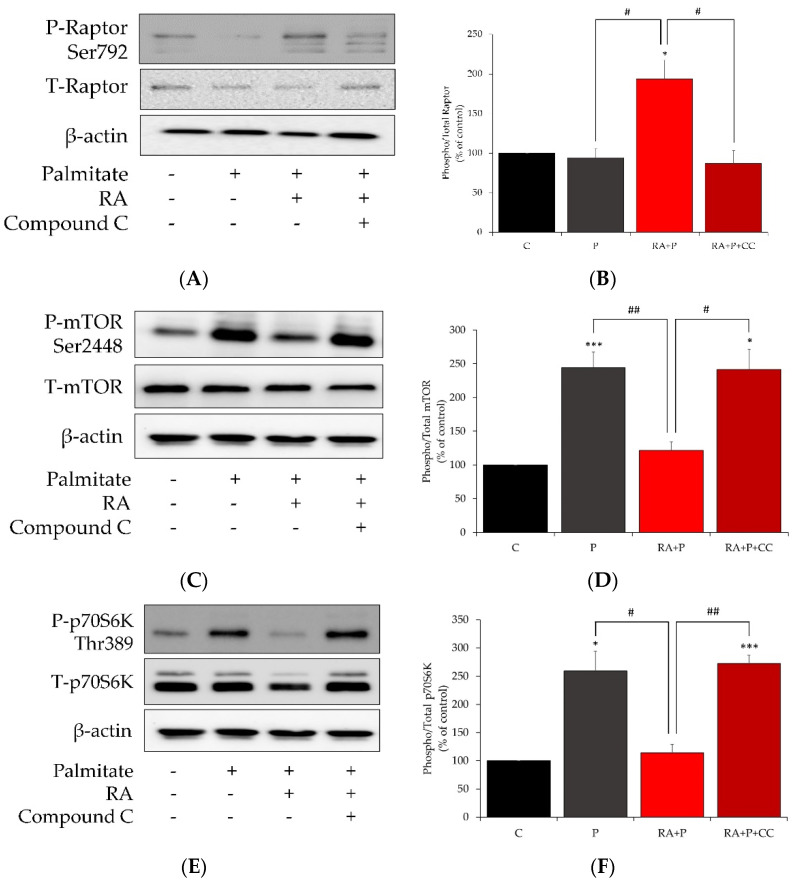
AMPK inhibition/compound C prevents the effects of rosmarinic acid on the phosphorylation of Raptor (**A**,**B**), mTOR (**C**,**D**), and p70S6K (**E**,**F**). Fully differentiated myotubes were treated without (control, C) or with 25 μM compound C (CC) for 30 min followed by treatment with 0.2 mM palmitate (P) for 16 h in the absence or the presence of 5 μM rosmarinic acid (RA). After treatment, the cells were lysed, and SDS-PAGE was performed, followed by immunoblotting with specific antibodies that recognize phosphorylated Ser^792^ or total Raptor, phosphorylated Ser^2448^ or total mTOR, and phosphorylated Thr^389^ or total p70S6K. Representative immunoblots are shown (**A**,**C**,**E**). The intensity of the bands (arbitrary units) was measured using Image J software and expressed as percent of control (**B**,**D**,**F**). The data are shown as the mean ± SE of three separate experiments (* *p* < 0.05, *** *p* < 0.001 vs. control, # *p* < 0.05, ## *p* < 0.01 vs. RA + P).

**Figure 8 ijms-24-05094-f008:**
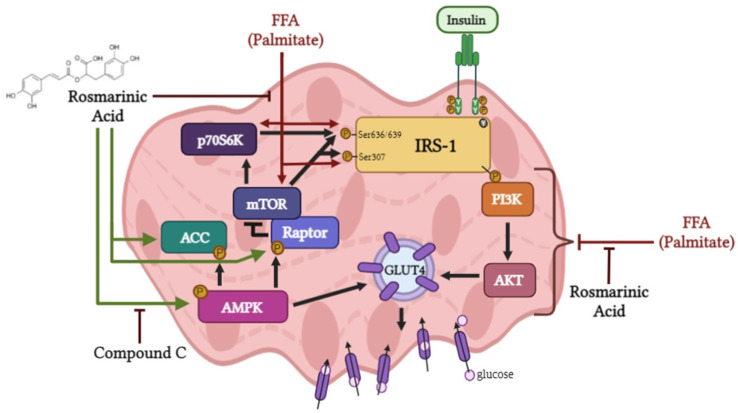
Rosmarinic acid counteracted the free fatty acid (FFA; palmitate)-induced muscle cell insulin resistance. Rosmarinic acid (RA) prevented the palmitate-induced phosphorylation/activation of mTOR and p70S6K, and increased the phosphorylation/activation of AMPK, ACC, and Raptor. RA restored insulin-stimulated Akt phosphorylation/activation, GLUT4 translocation, and glucose uptake in the presence of palmitate. Compound C (CC) prevented the effects of RA on the palmitate-induced phosphorylation/activation of mTOR and p70S6K and reduced the phosphorylation/activation of Raptor.

## Data Availability

Data available on request.

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
