# Peer review of "Muscle Cell Insulin Resistance Is Attenuated by Rosmarinic Acid: Elucidating the Mechanisms Involved"

_ijms, 2023, doi:10.3390/ijms24065094_

Round 1

Reviewer 1 Report

This is yet another insulin tolerance and AMPK study conducted by the authors on the components of rosemary. Too many natural products can affect the AMPK pathway, but few take actual pharmacological effects on T2DM or obesity. AMPK is a common inflammatory pathway associated with inflammation, metabolic homeostasis, cancer, and aging. Although the authors tested the regulations of RA on several proteins in the AMPK pathway in this manuscript, the pharmacological effects are still unclear. It is hasty and meaningless to carry out mechanistic exploration before the bio functions are clarified. The authors have already published six articles (ref 37 - 42) on the AMPK agonistic function of components in rosemary, and it seems that every composition in rosemary can take a try on AMPK, but it is very ambiguous whether this regulation can cause macroscopic biological activity. Can the authors positively elucidate the critical component of rosemary on obesity or glucose uptake? Has this key component been acquired by bio-guided isolation? Has the chemical structure of RA in this manuscript been identified by NMR? I think it is unacceptable and meaningless to carry out the pathway or mechanism exploration of AMPK for every molecule obtained and ignore the pharmacological effects.

Author Response

Reviewer 1  

This is yet another insulin tolerance and AMPK study conducted by the authors on the components of rosemary. Too many natural products can affect the AMPK pathway, but few take actual pharmacological effects on T2DM or obesity. AMPK is a common inflammatory pathway associated with inflammation, metabolic homeostasis, cancer, and aging. Although the authors tested the regulations of RA on several proteins in the AMPK pathway in this manuscript, the pharmacological effects are still unclear. It is hasty and meaningless to carry out mechanistic exploration before the bio functions are clarified. The authors have already published six articles (ref 37 - 42) on the AMPK agonistic function of components in rosemary, and it seems that every composition in rosemary can take a try on AMPK, but it is very ambiguous whether this regulation can cause macroscopic biological activity. Can the authors positively elucidate the critical component of rosemary on obesity or glucose uptake? Has this key component been acquired by bio-guided isolation? Has the chemical structure of RA in this manuscript been identified by NMR? I think it is unacceptable and meaningless to carry out the pathway or mechanism exploration of AMPK for every molecule obtained and ignore the pharmacological effects.

We thank the reviewer for the comment, and we appreciate the critique.

The rosmarinic acid (RA) used in the current study was purchased from Sigma Life Sciences (ST. Louis, MO, USA) CAT#4033.  The product data sheet states that it is extracted from Rosemarinus officinalis L and has purity of ≥98% (HPLC).

Our sole focus here was to examine if RA can counteract the deleterious effects of palmitate in muscle cells. Additionally, we wanted to examine if RA can restore insulin signaling and insulin responsiveness that is negatively impacted by palmitate.

Studies examining the effects of RA in metabolically active tissues such as liver, fat and muscle are very limited. Currently, there are only 2 studies that examine the effect of RA in free fatty acid-induced insulin resistant cells.  Jayanthy et al., 2017 investigated the effects of RA in palmitate-treated muscle cells however they did not examine the effects of RA in insulin-responsiveness in the presence of palmitate [1]. Another study examined the effects of RA in oleic-acid-induced hepatic steatosis in HepG2 liver cells (Balachander et. al., 2018) [2].   Therefore, only one other study exist today utilizing muscle cells in vitro[1] and this study contains major limitations as it did not examine insulin responsiveness.

In general, in vitro studies are the first step to assess specific biological effects of a chemical with a potential to treat diseases (in this case diabetes). We used L6 cells an established model of skeletal muscle that is widely used for insulin signaling studies since they express the insulin sensitive GLUT4.

We are examining biofunctions of RA. Our study is the first to examine signaling, mechanistic events affected by RA treatment.

All drug development starts with in vitro, mechanistic studies screening for potential biological targets. Such studies are not meaningless, they are meaningful.

Based on the above, we believe our data/study contributes towards better understanding of the antidiabetic potential of RA and deserves to be published.

References:

  1. Jayanthy, G.; Roshana Devi, V.; Ilango, K.; Subramanian, S.P. Rosmarinic Acid Mediates Mitochondrial Biogenesis in Insulin Resistant Skeletal Muscle Through Activation of AMPK. J. Cell. Biochem. 2017, 118, 1839–1848, doi:10.1002/jcb.25869.
  2. Balachander, G.J.; Subramanian, S.; Ilango, K. Rosmarinic Acid Attenuates Hepatic Steatosis by Modulating ER Stress and Autophagy in Oleic Acid-Induced HepG2 Cells. RSC Adv 2018, 8, 26656–26663, doi:10.1039/c8ra02849d.

Reviewer 2 Report

Review of Manuscript ID: ijms-2051538

Title: Muscle cell insulin resistance is attenuated by rosmarinic acid: Elucidating the mechanisms involved

Authors: Danja J. Den Hartogh, Filip Vlavcheski, And Evangelia Tsiani

General comment: the study reports the protective effect of rosmarinic acid in cultured myocytes made insulin resistant by treatment with palmitate. The authors use a cell culture model of insulin-resistant muscle cells, to show that the phenolic compound recovers the inhibited insulin-stimulated Akt phosphorylation and the insulin-stimulated glucose uptake and GLUT4 translocation as well as the increased phosphorylation of mTOR, and p70S6K induced by palmitate. The hypothesis and objectives are sound, methods seem adequate and results are interesting and with a translational potential in vivo due to the low realistic dose of rosmarinic acid. Although the study adds novel data regarding the molecular mechanisms involved in the anti-diabetic effect of a specific compound within a Rosmarinic officinalis extract, the fact is that most information is confirmation of previous works from the same group in the same experimental model, primarily with a Rosemary extract (ref. 41) and then with one extract compound, carnosic acid (ref. 42). Additionally, the anti-diabetic effect of rosmarinic acid has been previously reported in the same cell line (ref. 44) and even in vivo in diabetic rats (refs. 43,44); indeed, in the latter reference, regulatory effect of rosmarinic acid on AMPK phosphorylation and GLUT4 translocation are specifically described. All the above greatly reduces the biomedical and pharmacological impact of the study. Some specific comments are detailed below:

Specific comments:

1)      Line 70; the authors should also include flavanols as anti-diabetic polyphenols and AMPK regulators; see references: Martín et al. (2016) Anti-diabetic actions of cocoa flavonoids. Mol. Nutr. Food Res. 60: 1756-69; and Martín et al. (2017) Protective effects of tea, red wine and cocoa in diabetes. Evidences from human studies. Food Chem. Toxicol. 109: 302-314.

2)      Figures 3-4; if PA itself seems not to affect basal p-AKT, GLUT-4 and glucose uptake (figures 2-4), the condition P + RA in assays depicted in figures 3 and 4 seems useless.

3)      If mTOR and p70S6K are major kinases implicated in serine phosphorylation of IRS-1, the authors should explain the reason why palmitate alone enhances mTOR, p70S6K phosphorylation/activation, and IRS-1 inactivation without evoking any effect on p-Akt (figure 2) and, presumably, GLUT4 and glucose uptake.

Author Response

Reviewer 2

General comment: the study reports the protective effect of rosmarinic acid in cultured myocytes made insulin resistant by treatment with palmitate. The authors use a cell culture model of insulin-resistant muscle cells, to show that the phenolic compound recovers the inhibited insulin-stimulated Akt phosphorylation and the insulin-stimulated glucose uptake and GLUT4 translocation as well as the increased phosphorylation of mTOR, and p70S6K induced by palmitate. The hypothesis and objectives are sound, methods seem adequate and results are interesting and with a translational potential in vivo due to the low realistic dose of rosmarinic acid. Although the study adds novel data regarding the molecular mechanisms involved in the anti-diabetic effect of a specific compound within a Rosmarinic officinalis extract, the fact is that most information is confirmation of previous works from the same group in the same experimental model, primarily with a Rosemary extract (ref. 41) and then with one extract compound, carnosic acid (ref. 42). Additionally, the anti-diabetic effect of rosmarinic acid has been previously reported in the same cell line (ref. 44) and even in vivo in diabetic rats (refs. 43,44); indeed, in the latter reference, regulatory effect of rosmarinic acid on AMPK phosphorylation and GLUT4 translocation are specifically described. All the above greatly reduces the biomedical and pharmacological impact of the study. Some specific comments are detailed below: 

Specific comments:

1)      Line 70; the authors should also include flavanols as anti-diabetic polyphenols and AMPK regulators; see references: Martín et al. (2016) Anti-diabetic actions of cocoa flavonoids. Mol. Nutr. Food Res. 60: 1756-69; and Martín et al. (2017) Protective effects of tea, red wine and cocoa in diabetes. Evidences from human studies. Food Chem. Toxicol. 109: 302-314.

We thank the reviewer for this suggestion.

Addressed.  Please see the revised manuscript (Lines 70-71) and below:

“AMPK activity is driven by muscle contraction/exercise [3] and stimulated by several compounds including the currently used anti-diabetic drugs metformin [4] and thiazolidineones [5], and  by various polyphenols / flavanols found  in tea [6], red wine, citrus fruit and cocoa [7] including resveratrol [8], naringenin [9], and cocoa flavanol [10] that result in increased skeletal muscle glucose uptake. Utilizing AMPK activators has gained increasing attention as novel pharmacological interventions for the prevention/treatment of T2DM and insulin resistance [3,11–13]. Chemicals found in plants/herbs that activate AMPK have attracted attention as potential anti-diabetic treatment options.”

2)      Figures 3-4; if PA itself seems not to affect basal p-AKT, GLUT-4 and glucose uptake (figures 2-4), the condition P + RA in assays depicted in figures 3 and 4 seems useless.

We thank the reviewer for this suggestion. We agree with your suggestion and have removed RA+P from Figures 3 and 4.  See the updated figures in the revised manuscript and below.

Figure 3.                                                                                    Figure 4.

3)      If mTOR and p70S6K are major kinases implicated in serine phosphorylation of IRS-1, the authors should explain the reason why palmitate alone enhances mTOR, p70S6K phosphorylation/activation, and IRS-1 inactivation without evoking any effect on p-Akt (figure 2) and, presumably, GLUT4 and glucose uptake.

We thank the reviewer for this comment. Please see our explanation below.

Palmitate alone enhances mTOR, p70S6K phosphorylation/activation, and these kinases phosphorylate IRS-1 on serine residues. These effects are happening at the basal level.  When a stimulus arrives, that is insulin binding to its receptor (insulin-stimulated conditions), the serine phosphorylation of IRS-1 acts as a shield preventing its full tyrosine phosphorylation and activation of the PI3K-Akt signaling downstream. The basal Akt phosphorylation, GLUT4 and glucose uptake are not affected by palmitate, but the insulin-stimulated responses are impacted. 

At the basal level the majority of GLUT4 is located intracellularly and Akt phosphorylation is very low. When insulin binds to its receptor (insulin-stimulated conditions), a robust phosphorylation/activation of Akt occurs that causes GLUT4 transporter translocation to plasma membrane. That is a response seen in healthy muscle cells. When palmitate is around because of the shield (serine phosphorylation of IRS-1) the insulin signaling is impacted.

 Similar observations have been reported by Olefsky’s group (Kruszynska et al)  

and others (Cazzolli et al, Sinha et al).

References:

Kruszynska, Y.T.; Worrall, D.S.; Ofrecio, J.; Frias, J.P.; Macaraeg, G.; Olefsky, J.M. Fatty Acid-Induced Insulin Resistance: Decreased Muscle PI3K Activation But Unchanged Akt Phosphorylation. The Journal of Clinical Endocrinology & Metabolism 2002, 87, 226–234, doi:10.1210/jcem.87.1.8187.

Cazzolli, R.; Carpenter, L.; Biden, T.J.; Schmitz-Peiffer, C. A Role for Protein Phosphatase 2A–Like Activity, but Not Atypical Protein Kinase Cζ, in the Inhibition of Protein Kinase B/Akt and Glycogen Synthesis by Palmitate. Diabetes 2001, 50, 2210–2218, doi:10.2337/diabetes.50.10.2210.

Sinha, S.; Perdomo, G.; Brown, N.F.; O’Doherty, R.M. Fatty Acid-Induced Insulin Resistance in L6 Myotubes Is Prevented by Inhibition of Activation and Nuclear Localization of Nuclear Factor ΚB*.

Round 2

Reviewer 1 Report

I know the workflow of drug development. Generally, cell experiments are only suitable for high-throughput screening or mechanism research when the efficacy is apparent. The AMPK pathway is associated with a variety of diseases or drug functions. My concern is how to prove whether the regulatory effect of RA on AMPK is confirmed relating to the pharmacological effects described in this manuscript. If this pharmacological effect is explored on cells, a separate and simple pharmacological experiment like Figure 4 is not enough. At least, gradient concentrations of RA need to be explored on insulin resistance. Dose-dependent manner can indicate that RA has the effect of improving insulin resistance in cells. But as a more rigorous exploration, it also needs to be verified in vivo.

Any exogenous substance will cause more or fewer responses of pathways in the body or cells. Especially omics experiments can see this clearly. As a manuscript for mechanism exploration, I think it is highly necessary to clarify the correlation between invoked pathways and biological functions. There must be sufficient experimental evidence on your cell lines to prove the effectiveness of RA.

So, I think additional pharmacological, or phenotype experiments are needed.

Author Response

Reviewer Comments

I know the workflow of drug development. Generally, cell experiments are only suitable for high-throughput screening or mechanism research when the efficacy is apparent. The AMPK pathway is associated with a variety of diseases or drug functions. My concern is how to prove whether the regulatory effect of RA on AMPK is confirmed relating to the pharmacological effects described in this manuscript. If this pharmacological effect is explored on cells, a separate and simple pharmacological experiment like Figure 4 is not enough. At least, gradient concentrations of RA need to be explored on insulin resistance. Dose-dependent manner can indicate that RA has the effect of improving insulin resistance in cells. But as a more rigorous exploration, it also needs to be verified in vivo.

Any exogenous substance will cause more or fewer responses of pathways in the body or cells. Especially omics experiments can see this clearly. As a manuscript for mechanism exploration, I think it is highly necessary to clarify the correlation between invoked pathways and biological functions. There must be sufficient experimental evidence on your cell lines to prove the effectiveness of RA.

So, I think additional pharmacological, or phenotype experiments are needed.

We thank the reviewer for the suggestions.

In the past we have performed dose-response experiments using 0.1, 0.5, 2, 5 and 10 µM of RA (see Figure 1 Vlavcheski et al Molecules 2017 [1]) and found that maximum stimulation of glucose uptake was achieved with 5 µM RA. In that study, we compared the effect of RA with that of insulin and metformin, the most widely used/prescribed medication for T2DM and we found that the maximum stimulation of glucose uptake seen with RA treatment (186 ± 7.31% of control, p < 0.001) was comparable to the response seen with maximum insulin (100 nM, 0.5 h, 204 ± 10.73% of control, p < 0.001) and metformin (2 mM, 2 h, 202 ± 14.37% of control, p < 0.001) stimulation (see Figure 3A, Vlavcheski et al Molecules 2017 [1])

 Based on these evidence/ past study, we did not perform dose-response experiments and utilized 5 µM RA in the present study. 

Following the reviewer’s suggestion, we have performed additional experiments utilizing Compound C, an AMPK inhibitor, in an attempt to elucidate the mechanism of action of RA. Please see the new data below and in Figure 7 of the revised manuscript.

2.7. AMPK inhibition reverses the effects of Rosmarinic acid on palmitate-induced phosphorylation of Raptor, mTOR, and p70S6K.

 In an attempt to elucidate the mechanism of action of RA, we performed additional experiments utilizing compound C (CC), a specific AMPK inhibitor. The phosphorylation of Raptor at Ser792 was significantly increased in cells treated with 5 µM RA in the presence of 0.2 mM palmitate (RA+P: 193.63 ± 23.7% of control, p<0.05, Figure 7A, B) and importantly, pretreatment of the cells with CC abolished this response (RA+P+CC: 87.2 ± 16.04% control, p<0.05, Figure 7A, B). Furthermore, in the presence of CC the effect of RA on suppressing the palmitate-induced mTOR and p70S6Kphosphorylation/activation (P: 244.2 ± 23.3% and 259.3 ± 34.8% of control, p< 0.001 and p<0.05, respectively, Figure 7C-F), (RA+P: 121.6 ± 12.8% and 114.1 ± 14.7% of control, p<0.01 and p<0.05, respectively, Figure 7C-F) was abolished (RA+P+CC: 241.5 ± 30.1% and 272.2 ± 14.9% control, p<0.05 and p<0.01, respectively, Figure 7C-F).

                                                                                                (A)                                                             (B)

                                             (C)                                                                 (D)

                                          (E)                                                                 (F)

Figure 7. AMPK inhibition/ compound C prevents the effects of rosmarinic acid on the phosphorylation of Raptor (A, B), mTOR (C, D), and p70S6K (E, F). Fully differentiated myotubes were treated without (control, C) or with 25 μM compound C (CC) for 30 min followed by treatment with 0.2 mM palmitate (P) for 16 hours in the absence or the presence of 5 μM rosmarinic acid (RA). After treatment, the cells were lysed, and SDS-PAGE was performed, followed by immunoblotting with specific antibodies that recognize phosphorylated Ser792 or total Raptor, phosphorylated Ser2448 or total mTOR, and phosphorylated Thr389 or total p70S6K. Representative immunoblots are shown (A, C, E). The intensity of the bands was measured and expressed in arbitrary units (B, D, F). The data is the mean ± SE of three separate experiment (*p<0.05, ***p<0.001 vs. control, #p<0.05, ##p<0.01 vs. RA+P).

  1. Vlavcheski, F.; Naimi, M.; Murphy, B.; Hudlicky, T.; Tsiani, E. Rosmarinic Acid, a Rosemary Extract Polyphenol, Increases Skeletal Muscle Cell Glucose Uptake and Activates AMPK. Molecules 2017, 22, doi:10.3390/molecules22101669.

Reviewer 2 Report

The authors have conveniently addressed all my comments and queries; thus, I recommend to accept the revised version 1 for publication at IJMS.

Author Response

Reviewer 2

The authors have conveniently addressed all my comments and queries; thus, I recommend to accept the revised version 1 for publication at IJMS. 

We thank the reviewer for their acceptance of the revised manuscript.

Round 3

Reviewer 1 Report

All the concerns were fully replied and discussed.

I have no further concerns.